# Acute Toxicity of Divalent Mercury Ion to *Anguilla japonica* from Seawater and Freshwater Aquaculture and Its Effects on Tissue Structure

**DOI:** 10.3390/ijerph16111965

**Published:** 2019-06-03

**Authors:** Yuanqiang Tang, Yunguo Liu, Tao Zhang, Jiang Li, Xiaohua Wang, Wei Zhang, Guangming Zeng, Shaobo Liu, Lei Guan

**Affiliations:** 1College of Environmental Science and Engineering, Hunan University, Changsha 410082, China; bobtang2016@hnu.edu.cn (Y.T.); wangxiaohua6999@126.com (X.W.); zhangweisx@hnu.edu.cn (W.Z.); 2Key Laboratory of Environmental Biology and Pollution Control, Hunan University, Ministry of Education, Changsha 410082, China; 3Meijiang County Aquatic Technology Promotion Station, Meizhou 514000, China; 13430112897@163.com; 4School of Architecture and Art, Central South University, Changsha 410082, China; lijiang@csu.edu.cn (J.L.); liushaobo23@aliyun.com (S.L.); 5School of Architecture and Urban Planning, Hunan City University, Yiyang 413000, China; 6Meizhou Fisheries Research Institute, Meizhou 514000, China; guanleia@126.com

**Keywords:** mercury, *Anguilla japonica*, acute toxic effects, safety concentration

## Abstract

The acute toxicity of divalent mercury ion to *Anguilla japonica* from seawater and freshwater aquaculture was assessed. In particular, the effects of toxicity on the microstructures of the gill and liver tissues were examined using the hydrostatic method, without feeding, at a water temperature of 20 °C. The median lethal concentrations (LC_50_) of divalent mercury ion to fishes in seawater and freshwater over various durations were: 24 h = 1.637 and 1.428 mg/L; 48 h = 1.562 and 1.377 mg/L; 72 h = 1.530 and 1.284 mg/L; and 96 h = 1.442 and 1.228 mg/L. The safety mass concentrations were 0.1442 and 0.01228 mg/L, respectively. After exposure to divalent mercury ion, adhesion between the gill lamellae and massive cellular disintegration and necrotic shedding were observed in the gill tissue sections. The liver tissues underwent hyperemia and swelling, with the appearance of blood spots, swelling of the hepatocyte mitochondria, dilation of the rough endoplasmic reticulum, and intercellular inflation.

## 1. Introduction

Rapid industrial and agricultural development, increased urban population size, and improvements to people’s living standards have caused large amounts of wastewater containing heavy metals to flow into rivers, lakes, and oceans. The pollution sources of heavy metals include volcanic activity, rock weathering, soil erosion and other natural factors, as well as anthropogenic factors such as train flue gas discharge, sewage irrigation, construction engineering, metal products manufacturing, mining, incineration, smelting and so on (see Figure 1) [1]. Mercury is the main heavy metal pollutant in many reservoirs and rivers in China. Chromium and lead pollution in drinking water sources of surface water is common [2]. The resultant pollution of the aquatic environment is seriously affecting the survival of fishes [3]. Under the action of microorganisms [4], the various forms of mercury present in water are converted to methylmercury [5], which enters fishes through the enrichment effect of the food chain [6,7]. This phenomenon is called bioaccumulation [8]. After these fish containing heavy metals are eaten by human beings, they will cause greater harm to human health after enrichment and magnification in the food chain.

According to the priority pollutants list published by USEPA [9], mercury, cadmium, chromium and zinc are priority pollutants in water environment. Most researchers agree that mercury is the most toxic element (Hg^2+^ >> Cd^2+^ > Zn^2+^ > Cr^+6^ >> Ni^2+^) [10,11,12,13]. The toxicity of methylmercury is several times that of inorganic mercury [14]. However, there is much controversy over whether inorganic mercury can be methylated inside fishes and converted to methylmercury [15,16,17]. Mercury is a highly toxic metal that is found in the spawning and nursery areas of red sea bream, and is believed to have deleterious effects on wild populations (see Figure 1).

Fishes inhabit in the water constantly, and are highly sensitive to different pollutants in the aquatic environment throughout the various stages in their lifespans. Hence, using fishes as test organisms for acute toxicity has the advantages of convenience, sensitivity, and rapidity [18]. The test fish species used in this study was the Japanese eel (*Anguilla japonica*), which belongs to the order Anguilliformes, family Anguillidae, and genus Anguilla. Snake-like, and small its life span is over 50 years, generally occurring in saltwater and freshwater waters. It is mainly distributed in the Yangtze River, Minjiang River, Pearl River Basin, Hainan Island and rivers and lakes of China (see Figure 2 and Figure 3). There are 18 species of eels in the world, including Japanese eel, perch eel, Celebes eel and short-fin eel in Taiwan, but Japanese eels are the most abundant, while the other three species are relatively scarce, because migratory fish around the world suffer from a variety of threats during migration. This situation is positively related to the pollution of the living environment, in addition to the heavy fishing pressure. Eels have existed on Earth for thousands of years. *Anguilla* grow in fresh water because of their peculiar growth process (eggs are laid in the marine environment) in order to cope with the environmental stress, the sex of female and male will change with the change of the environment.

Therefore, understanding fish’s response to heavy metal ions in the natural environment from a physiological perspective may clarify the vulnerability of fish and the in situ [20,21,22,23,24] repair capacity of the fishery environment [25]. These patterns may be particularly important for threatened species, as management depends largely on the ability of research groups to determine how major stressors directly (e.g., mortality) or indirectly (e.g., growth and/or health disorders) affect their populations. Additionally, *Anguilla* is a suitable organism for testing toxicity to fishes in the laboratory because it is easy to breed, with strong mobility, and early sexual maturity.

However, in recent years, due to the rise of the animal protection movement, the voice of animal rights protectionist has become louder and louder, and animal experiments are facing severe ethical challenges. Emerging topics, such as how to make animal experiments meet the requirements of bioethics? What is the relationship between bioethics and contemporary biology, environmental health and medical ethics [26,27]? How to safeguard the welfare ethics of experimental animals have become a hot issue of concern to the deep-gray public, governments and scientists, and has influenced research trends, the sociality of results and the public recognition of the subjects related to animal experiments. The interests of animals can serve as the basis for animal rights, such as the interests of avoiding pain in order to avoid the responsibility of spasmodic pain and the rights associated with freedom from pain. Like human rights, there are sometimes good reasons to go beyond animal rights. Human beings have the right to life, but this right is reasonably overthrown in justice. Animals have the right to enjoy a painless life, but this right is reasonably covered in important biomedical research. Part of the reason why humans have reason to overwhelm animal rights is to weigh interests, which is also the focus of animal ethics research. All this has nothing to do with the topic of this paper with the rapid development of biomedicine and food pharmacy research, the importance of laboratory animals in scientific research has become increasingly prominent. Prevention, diagnosis and treatment of diseases, safety evaluation of drugs and devices, and major research closely related to human living environment, all need the support of laboratory animal work to complete. There is no doubt that laboratory animals play an important role in protecting human health and optimizing living environment. Animal experiment is still an indispensable means in the fields of biology, environment and medicine. In a word, it is not only rats used in experiments, but also fish in different culturing environments. The former is more important than protecting human health and avoiding animal pain or loss of life. This is different from the general view of equal interests.

Although studies on heavy metal residues in *Anguilla japonica* exist [28], there is no report on the toxicity of heavy metal to the fish and effects on its histology. In this study, since eels are migratory fish in rivers (originating from the ocean, growing in fresh water and returning to the ocean to spawn) the hydrostatic method was used to test the median lethal concentrations of solutions of heavy metal mercury to *A. japonica*. In parallel, the effects of this substance on the gill and liver tissues were evaluated. This study aimed to evaluate the effects of heavy metals on the survival and growth of this aquatic organism. The purpose of this study is to find the safe and lethal concentration of eel through sub-lethal tests, which prove that mercury ion has an important impact on eel culture and production, and the findings are expected to provide a basis for the Chinese Ministry of Environmental Protection to formulate standards for water quality and wastewater discharge, and for the Chinese Ministry of Agriculture’s Bureau of Fisheries to propose measures for pollution prevention and the protection of water environmental resources.

## 2. Materials and Method

### 2.1. Test Animals, Experimental Conditions, and Toxic Reagent

The test fishes were *A. japonica* bred in concrete ponds of seawater and freshwater. In March 2015, Meizhou Fisheries Station of Guangdong Province introduced 20,000 eel fry from Qidong County, Nantong City, Jiangsu Province. The fry was healthy and vigorous, bright in color, sensitive in response, with a body length of 15 cm and a body diameter of 1 cm. After more than two years of feeding, the eel reached 40 cm and its body diameter reached 2.5–3 cm. In April 2017, experimental fish were selected from the batch, according to the following inclusion criteria: the fish, which with large size, sensitive reactions and normal color, were selected as experimental ones. Individuals had an average body length of 25 cm. The quality of the pond water was in line with the national breeding standards for seawater and freshwater aquaculture. The fishes were caught and kept in the laboratory aquarium for 1 day. Fishes with normal motility, good health, responsiveness, and similar specifications were screened and randomly divided into groups of 10 individuals for the experiment. The experiment was carried out at the Meijiang County Aquatic Technology Promotion Station in Meizhou City, China. The fishes were reared in an aquarium of 50 × 30 × 28 cm size. The water used in the experiment was fully aerated. Each tank contained 20 L water that was maintained at 20 °C. Analytically pure grade HgSO_4_ was provided by Guizhou Tongren Yinhu Chemical Co., Ltd., Tongren City, China. Stock solution was prepared in advance and diluted to the various mass concentrations according to the needs of the experiment.

### 2.2. Acute Toxicity Experiment

The hydrostatic biological test method was used during the experiment [29]. The test solution was not replaced during the experimental process, but was continuously aerated throughout the day. The fishes were not fed during the experimental period to eliminate any effect that the feed might cause. A preliminary test was carried out to determine the 100% lethal and maximum tolerated concentrations of the heavy metal divalent mercury ion. Based on the results of the preliminary test, 20 treatment groups were set up: nine experimental groups were set up in seawater and freshwater, respectively, with different mercury ion concentrations, and two control groups were set up. Each group contained 10 fishes. The behavior, poisoning symptoms, and mortality rates of fishes were observed during the exposure process (see Table 1). When an individual fish showed no reaction to multiple stimuli, it was deemed dead and was removed from the water in a timely manner (before 6 a.m. or 6 p.m.). The number of dead fishes in the various treatment groups at 24, 48, 72, and 96 h was recorded, and the average mortality rate of each experimental group was calculated.

### 2.3. Preparation of Tissue Sections and Data Processing

The gills and livers of poisoned and normal (control) fishes were removed from the bodies, fixed in Bouin’s fluid for 3 h, washed and dehydrated to transparency, and then embedded in paraffin. Flakes of 7 μm thickness were generated by way of using a slicer and placed onto patches before staining them with the H·E method. After dehydration to transparency, the sections were sealed with a neutral gum and sliced. A Samsung SCB-1000 (Tianjin Samsung Opto-Electronics Co., Ltd., Tianjing, China) was used for observations and photography. The results of the acute toxicity experiment of divalent mercury ion to *A. japonica* specimens were processed with the probability unit–logarithmic plot method. Next, the median lethal mass concentration LC_50_ of the heavy metal ion to the fishes after 24, 48, 72, and 96 h of exposure was obtained. The following formula was used to calculate the safety mass concentrations [30]:Safety concentration (SCI) = LC_50_ (96 h) × 0.01

## 3. Results and Analysis

### 3.1. Acute Toxicity of Divalent Mercury Ion to Anguilla japonica

In the test aquarium, various reactions of the fishes to poisoning were observed at different Hg^2+^ concentrations. However, the reactions of *A. japonica* to mercury in the two types of aquaculture were basically similar under the same conditions. The motility of the fishes in the low mass concentration groups was similar to that in the control groups. Motility within 48 h was almost unchanged, with most fishes swimming slowly in the water or at the bottom of the test aquarium.

The toxic reaction in the high mass concentration groups was rapid, with some test fishes dying within 24 h. The fishes appeared agitated initially, swimming around rapidly in the aquarium or darting up and down. Some individuals turned sideways or rolled around. Such behaviors persisted for several hours, after which some became more lethargic when swimming. The fish gradually lost their ability to move and became less responsive to external stimuli. Eventually, they stopped swimming and laid down at the bottom of the aquarium. The poisoned individuals had additional mucus on the surface and either laid flat or with the abdomen facing upwards after dying. The bodies became white in color and hardened. In contrast, there was zero death in the control groups.

The toxicity of mercury ion to *A. japonica* and the mortality rates rose with increasing concentrations and longer experimental durations. The median lethal and safe concentrations of divalent mercury ion to *A. japonica* in seawater and freshwater at 24, 48, 72, and 96 h were obtained from statistical analysis of the data (Table 2). Hg clearly affected *A. japonica* differently in seawater versus freshwater. Specifically, divalent mercury ion had slightly stronger toxicity on eels in seawater, while eels in freshwater exhibited higher tolerance.

### 3.2. Histological Changes to the Gills and Liver of Anguilla japonica

The gills of fishes are not just respiratory organs, but also contribute to other physiological activities, such as the excretion of metabolites, osmotic balancing of body fluids, and regulation of acid–base balance. The gills are in direct contact with the aquatic environment, and are the main target organ exposed to the toxins of chemical pollutants. The structural and physiological changes of the gills effectively indicate the level of contamination in the water body, and directly reflect the toxicity of chemicals to fishes [10,31,32,33]. Damage to gill tissue cells after exposure to toxic substances has been classified into two types: (1) damage caused by defensive reactions, and (2) direct damage. The former includes hypertrophy and hyperplasia of the epithelial cells in the gill filaments and edema of the respiratory epithelium in the gill lamellae. The latter includes necrosis and shedding of epithelial cells [34,35].

The gill filaments and lamellae of fishes in the control group were structurally intact and arranged in a dense and orderly manner. The gills of fishes in the experimental groups were in direct contact with the poisonous solution and were susceptible to damage. After being poisoned, the following changes were observed to the gill tissues: the gill lamellae became thinner and there was cellular necrosis and shedding (Figure 4a–d). After mercury ion exposure, the gill tissues of fishes in seawater underwent the following changes: curling of the gill filaments, adhesion between the gill lamellae, and massive cellular disintegration and necrotic shedding (Figure 4c). In contrast, the gill tissues of fishes in freshwater underwent cell proliferation and apical enlargement, had non-tissue cavities in the gill filaments, and hyperemia in localized areas (Figure 4d).

The liver is an important organ in the body of organisms and plays multiple important roles. These roles include detoxification, metabolic function, bile secretion, and immune defense. Normal liver tissue structures were observed in fishes from the control group. Specifically, the livers were glandular, with extremely high numbers of cell cords densely packed in netlike formations. The hepatic cell cords were polygonal, with varying sizes, but did not have any morphological mutations. Sinusoids were observed between the cell cords, some of which were filled with blood cells (Figure 4e,f). After mercury exposure, the liver tissues underwent hyperemia and swelling, and blood spots appeared. The mitochondria in hepatocytes also swelled, the rough endoplasmic reticulum became dilated, and intercellular inflation occurred. Eventually, liver necrosis led to the death of the fishes (Figure 4g,h).

## 4. Discussion

### 4.1. Toxicity and Median Lethal and Safety Concentrations of Divalent Mercury Ion to Anguilla japonica

The experiment showed that, for the same duration of exposure, the mortality rate of *A. japonica* rose with increasing Hg^2+^ concentrations in water. Thus, within a specific concentration range, divalent mercury ion has acute toxicity on this species. At the same concentration, the mortality rate increased with prolonged exposure to mercury poisoning, indicating that it accumulates within the body.

There was a noticeable difference in the toxicity of divalent mercury ion to *A. japonica* in seawater versus freshwater. The safety concentrations of mercury to seawater and freshwater were 0.01442 and 0.01228 mg/L, respectively. Many previous studies showed that mercury is toxic to fishes and shrimps. However, the safety concentration of divalent mercury ion to *A. japonica* measured in this experiment (0.01442 mg/L) was much higher than that recorded for the postlarvae of *Litopenaeus vannamei* (0.0021 mg/L) [36] and the postlarvae of *Marsupenaeus japonicus* (0.0012 mg/L) [37]. However, the safety concentrations of the current study were much lower than that recorded for *Carassius auratus auratus* (0.023 mg/L) [38], and the postlarvae of *Misgurnus anguillicaudatus* (0.071 mg/L) [39]. The current study also found that the toxicity of mercury to *A. japonica* in seawater was higher than that in freshwater. Thus, the toxicity of mercury to *A. japonica* varies depended on the living environment. This is because the toxicity of mercury ion is influenced by physicochemical factors, such as salinity, dissolved oxygen, and pH, which then affects the LC_50_ value. Therefore, the effects of mutual interactions between the various physicochemical factors in the living environment of fishes should be considered during practical applications. In the natural environment, organisms living in chronically polluted sites are exposed to low concentrations of metals for long periods. In other cases, organisms might be abruptly exposed to high levels of metals upon the outfall of a pollutant in coastal waters.

### 4.2. Effects of Divalent Mercury Ion on the Structure of the Gill and Liver Tissue of Anguilla japonica

The gills are the main respiratory organ of fishes. Their unique structure allows ions to pass through from the water. As a result, the gills are the main part of the fish body that absorb heavy metals directly from the water. The gills of *A. japonica* individuals are an important organ for gaseous exchange and the regulation of ion balance in the water. With increasing mercury concentration in the gill tissues and longer experimental durations, the tissues underwent various degrees of damage. The large amount of mucus that was secreted by the fish damaged the functioning of the gills, resulting in respiratory difficulties. For the control group, the gills included the visceral arch and gill filament, with many gill lamellae on each filament. The lamellae were arranged neatly and consisted of epithelial tissues, connective tissues, capillaries, and nerve tissues. Longitudinal sections of the lamellae revealed the capillaries within, which were filled with blood cells.

The experimental results showed that varying mercury concentrations caused different degrees of damage to the gill tissues. When the mercury concentration in the waterbody was low, the gill tissues did not undergo pathological changes. When the mercury concentration was ≥10 μg/L, the mucus secreted by the gill tissues gradually increased, particularly over longer experimental durations. The gill lamellae curled and the epithelium swelled. The lamellae also had apical enlargement, due to hyperemia, and became spherical or rod-shaped. Cells were shed from some lamellae, to the extent of tissue disintegration. This damage seriously affected the normal functioning of gaseous exchange by the gills. Guan et al. [40] showed that mercury in the bodies of fish was converted to methylmercury through the process of methylation. This organic compound invaded the nervous system of *A. japonica*, and severely damaged the gill tissues. There are adhesions between the gill lamellae, as well as massive cellular disintegration and necrotic shedding [40].

This study demonstrated that the heavy metal mercury ion increased vacuole formation in hepatocytes, with hepatic disintegration occurring in severe cases. The degree of damage increased with prolonged exposures to mercury, indicating that it accumulates within the body. This finding supports that of Kuang et al. [28], who identified the residual roles of heavy metals ion inside *A. japonica*.

Various research results in different regions of a number of countries have shown that mercury concentrations in the blood and hair of humans are related to eating fish [41]. In particular, Urieta et al. [42] showed that a population of adult Spaniards absorb an average of 18 pg of mercury per day in their dietary intake, of which 80–90% come from fish. Sanzo et al. [43] showed that about 95% of mercury in the diet of people comes from the consumption of fish and shellfish. Therefore, eating fish is the main source of mercury intake in the diet of the general population.

Toxicity data are fundamental for evaluating the potential impact of pollution on marine ecosystems. Environmental conditions are also important, impacting metal bioavailability and bioaccumulation levels. Toxicological bioassays involving a series of acute toxicity tests at different concentrations were used to evaluate the effects of heavy metals on *A. japonica* after 48 and 96 h in the current study. Standard biotoxicity tests usually determine cell growth and viability in relation to the pollutants of concern. Such tests are especially important for regulation purposes, which aim to prevent possible damage to biota and humans by establishing maximum tolerable levels of toxicants.

Heavy metals are also a threat to aquatic environments because they can be converted from inorganic to organic forms through biological action. Metallo-organic compounds, such as methylmercury, are highly toxic, and easily penetrate biological membranes to be fixed in the tissues of organisms [41]. In fact, membrane injury is one important effect of metal ions that may lead to the disruption of cellular functions. The data collected in the current study reinforce the observation that, to assess potentially dangerous compounds in a satisfactory manner, many ecologically different species must classified.

## 5. Conclusions

In this paper, the biological test of mercury sulfate solution was carried out, which provided some valuable conclusion for establishing the maximum tolerance level of mercury ion in eel:The design method, process and results used in this experiment will provide some reference for the follow-up eel experiment.This paper has a certain reference significance for the staff engaged in aquaculture, scientific research and other fields in the treatment of aquaculture wastewater.This paper has a certain reference value for researchers engaged in the field of environment and food safety and managers of relevant government functional departments when they study, deal with unexpected environmental disasters and make emergency decision-making deployment.

Moreover, from the point of view of watershed environmental protection, the process and results of this experiment will be helpful for the accurate determination of mercury concentration in aquatic environments and biological samples by chemical analysis methods in the future, and provide valuable theoretical support and data reference for government departments to formulate mercury control as one of the priority pollutants in aquatic environment.

## Figures and Tables

**Figure 1 ijerph-16-01965-f001:**
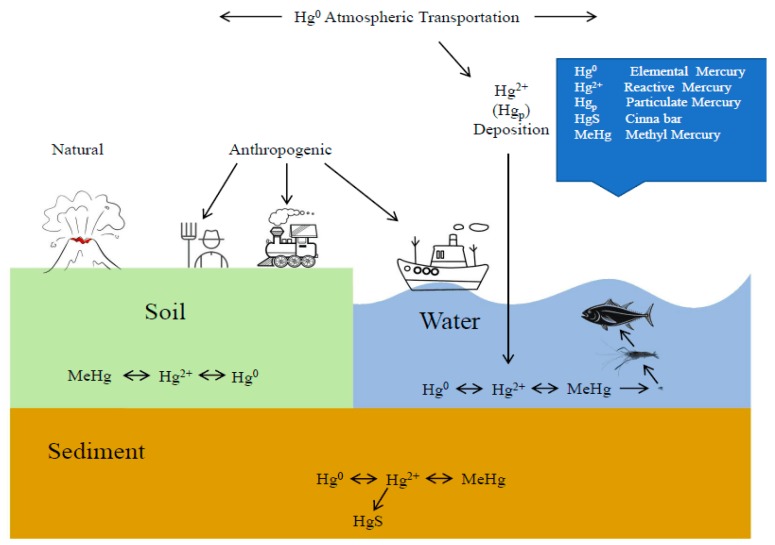
The global mercury cycle as adapted from Lindsay D. Starr [1]. The mercury cycle shows the transportation of Hg through the atmosphere, water, soil, and sediment.

**Figure 2 ijerph-16-01965-f002:**
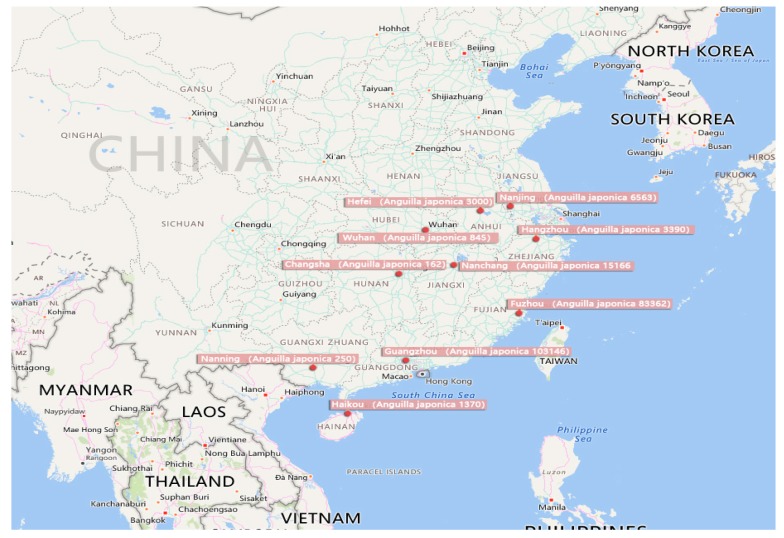
Distribution map of eel production in China (the output of eel culture in various provinces is taken from the 2018 China Fishery Statistics Yearbook [19], and the unit omitted after the number on the way is ton.

**Figure 3 ijerph-16-01965-f003:**
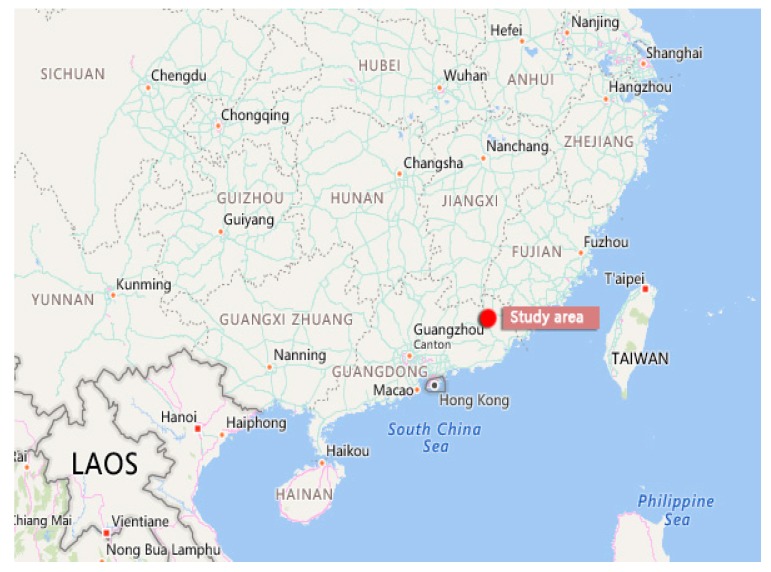
Study area (Meizhou city, Guangdong Province, China).

**Figure 4 ijerph-16-01965-f004:**
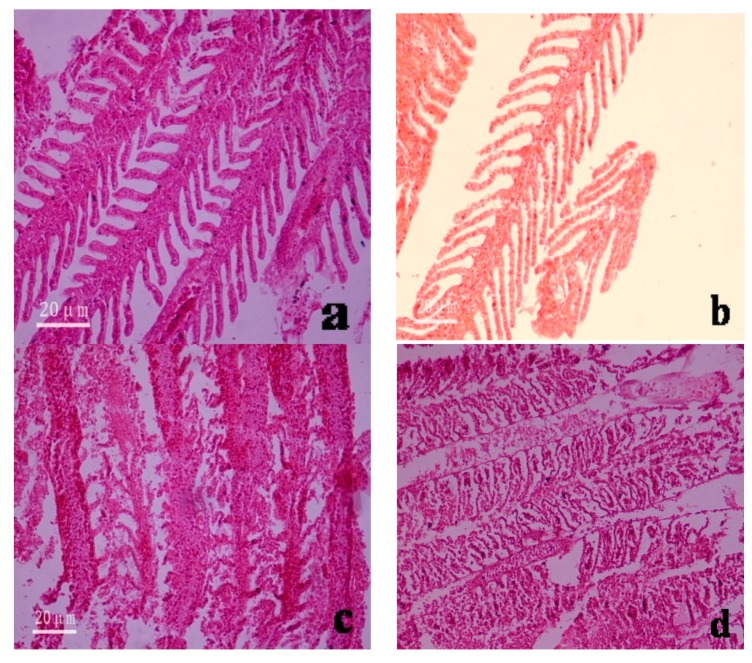
Microstructure of the gill and liver tissues of *Anguilla japonica* under the microscope (×200). (**a**) Normal gill tissues (seawater); (**b**) Normal gill tissues (freshwater); (**c**) Gill lamellae after divalent mercury ion treatment (seawater); (**d**) Gill lamellae after divalent mercury ion treatment (freshwater); (**e**) Normal liver tissues (seawater); (**f**) Normal liver tissues (freshwater); (**g**) Inflation between hepatocytes after divalent mercury ion treatment (seawater); (**h**) Appearance of blood spots among liver tissues after divalent mercury ion treatment (freshwater).

**Table 1 ijerph-16-01965-t001:** Divalent mercury ion mass concentrations and mortality rates of *Anguilla japonica.*

Hg^2+^ Mass Concentration	Mortality Rate (%)			
	24 h	48 h	72 h	96 h
Seawater				
0.891250938	0	0	0	0
1	0	0	0	10
1.122018454	10	20	20	20
1.258925412	10	20	20	30
1.412537545	30	30	40	50
1.584893192	50	50	60	60
1.77827941	50	60	60	70
1.995262315	70	80	80	90
2.238721139	100	100	100	100
Freshwater				
0.891250938	0	0	0	0
1	0	10	20	20
1.122018454	20	30	30	40
1.258925412	30	30	50	50
1.412537545	50	50	60	70
1.584893192	60	70	70	90
1.77827941	80	80	100	100
1.995262315	100	100	100	100
2.238721139	100	100	100	100

**Table 2 ijerph-16-01965-t002:** Median lethal and safety concentrations of divalent mercury ion to *Anguilla japonica*.

Hg^2+^	Experiment Duration	Regression Equation for Probability Unit—Concentration Logarithm	LC_50_	Safety Concentration
	(h)		(mg/L)	
Seawater	24	y = 10.228 × −2.190	1.637	0.01442
	48	y = 9.640 × −1.867	1.562	
	72	y = 9.557 × −1.765	1.530	
	96	y = 9.213 × −1.464	1.442	
Freshwater	24	y = 11.523 × −1.782	1.428	0.01228
	48	y = 10.112 × −1.404	1.377	
	72	y = 10.733 × −1.165	1.284	
	96	y = 12.231 × −1.091	1.228	

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
