# Peer review of "Acute Toxicity of Divalent Mercury Ion to Anguilla japonica from Seawater and Freshwater Aquaculture and Its Effects on Tissue Structure"

_ijerph, 2019, doi:10.3390/ijerph16111965_

Round 1
Reviewer 1 Report
Dear author,
The manuscript titled ”Acute toxicity of heavy metal Hg2+ to Anguilla japonica from seawater and freshwater aquaculture 3 and its effects on tissue structure Research ethics session for the use of animals is missing” is interesting to the reader giving some valuable results for biological tests aiming establishing maximum tolerable levels of Hg ions in Anguilla species. My only criticism is to the experimental session. By using metal solutions in measurements with biological tests, it is helpful to determine exactly the Hg concentration levels in the water as well as in the biota samples using chemical analysis methods. The proposed method of experimental design will give useful information about total Hg contents in Anguilla species and water samples used before and after the experiments.
Moreover, research ethics paragraph for the use of animals in research is missing.
I should propose for the publishing of the above manuscript after these minor revisions.
Author Response
We are very grateful to the valuable comments and suggestions from you and all the reviewers on our paper entitled “Acute toxicity of heavy metal divalent mercury ion to Anguilla japonica from seawater and freshwater aquaculture and its effects on tissue structure ” (Manuscript Number: ijerph-503772-revision). It is very helpful for us to make improvement. We have read and considered all the comments and recommendations carefully, and tried our best to revise the manuscript according to the comments of editors and reviewers.
The responses to each comment were marked in blue, and all of the modifications in our revised manuscript were marked in red, so that they could be easily identified. In the following pages, we outline each change made (point by point) as raised in the comments.
We hope you will be satisfied with our responses to the ‘comments’ and the revisions for the original manuscript. If you still have any question about this paper, please inform us and we will not hesitate one minute to modify.
Thank you very much for your thorough comments and constructive suggestions!

Reviewer 2 Report
Review ijerph-503772
Title: Acute toxicity of heavy metal Hg2+ to Anguilla japonica from seawater and freshwater aquaculture and its effects on tissue structure
Comments from reviewer
Overall, it is an interesting paper to test the median lethal concentrations of A. japonica to solutions of heavy metal mercury, as well as to evaluate the effects of this substance on the gill and liver tissues. However, I have some major and minor comments regarding the contents of your paper. Especially, some of sections (e.g. Introduction and study area in method section) need to be created and significantly expanded to support your scientific claims. Detail explanations are listed in below:
1. Introduction section is too short and less detail to describe the background of your research and to support your claims. Clearly, two paragraphs are clearly not enough to fully introduce and describe on your research. Thus, I highly urge you to expand the content of the introduction to fully delineate the background of the research and study objectives. For example, I can see that you have some references you cited in the introduction, so you can expand each sentence into greater details to make them more specific. Otherwise, you can add new references to introduce more extents in the introduction.
2. In the introduction, please clearly specify / list the objective of your study since it is little vague.
3. In the materials and method, you’ll need to include a section for study area with a map. As I read this paper, I didn’t see any information on where the fishes were caught and how 20 groups were collected in the study area. Please note that it’s very important to provide information on the study area, so that readers can get more reliable information on how well the study was conducted.
4. In the materials and method, subsections 2.1 , 2.2, and 2.3 are too short to be a single subsection. I’d suggest you to merge 2.1, 2.2, and 2.3 as one subsection and rename it as 2.1 Test animals, experimental conditions, and toxic reagent. Otherwise, I’d suggest you to expand each subsection with more detail information.
5. Same comment as #3 above, subsection 2.5 and 2.6 also need to be little more expanded. Otherwise, consider of merging them together.
6. Conclusion should be included in the paper as a separate section or with discussion section. Conclusion should be written in a way to summarize and conclude your overall findings and implementations.
Author Response

(The authors gave the same response as above.)

Reviewer 3 Report
Consider Mercury in the title not Hg2+
Do not use the term Heavy Metal anymore. Check
Pourret, O., 2018. On the Necessity of Banning the Term “Heavy Metal” from the Scientific Literature. Sustainability, 10(8): 2879.
Do not recall words from the title in the keywords list.
At the beginning of a sentence write Mercury not Hg.
Authors need to expand their litterature review. Check
Sonke, J.E., Heimbürger, L.-E., Dommergue, A., 2013. Mercury biogeochemistry: Paradigm shifts, outstanding issues and research needs. Comptes Rendus Geoscience, 345(5): 213-224.
Alpers, C.N., 2018. Mercury. In: White, W.M. (Ed.), Encyclopedia of Geochemistry: A Comprehensive Reference Source on the Chemistry of the Earth. Springer International Publishing, Cham, pp. 895-900.
Authors need to justify the list of most toxic elements, where does it come from? Blacksmith Institute? Pb, Hg, Cr, Cd?
Authors can make reference to Minamata Disease in the introduction, check
Harada, M., 1995. Minamata Disease: Methylmercury Poisoning in Japan Caused by Environmental Pollution. Critical Reviews in Toxicology, 25(1): 1-24.
Why considering Hg2+ and not Hg0?
L34 this phenomenon is called biomagnification. Check Lavoie, R.A., Jardine, T.D., Chumchal, M.M., Kidd, K.A., Campbell, L.M., 2013. Biomagnification of Mercury in Aquatic Food Webs: A Worldwide Meta-Analysis. Environmental Science & Technology, 47(23): 13385-13394.
Figure 1 is not of good quality due to distortion and not as precise as needed. Not the good reference!
Authors need to give details on Hg analysis, standard, QA/QC...
Authors used HgSO4, not Hg2+! Authors really need to discuss Hg speciation, give pH and electrolyte background. May be add a speciation modeling calculation considering the system chemistry.
Author Response

(The authors gave the same response as above.)

Round 2
Reviewer 2 Report
Thanks for addressing my comments in the paper. I can clearly see that your paper has been improved by addressing all reviewers' comments. However, I still have some minor comments on your paper. They are as follows:
Your study area is added in the paper, but all the labels in Figures 2a and b are shown in Chinese. Please consider this paper will be read by all around counties, not only Chinese. Please recreate your map figures written in English, so that they are readable to all audiences.
In conclusion, you don't need to list 5.1, 5.2 etc. Instead, please summarize main points of your paper, so that readers can get the primary information / findings from the conclusions.
Author Response
1. Your study area is added in the paper, but all the labels in Figures 2a and b are shown in Chinese. Please consider this paper will be read by all around counties, not only Chinese. Please recreate your map figures written in English, so that they are readable to all audiences.
Response: Thank you for your good advice. Following your suggestion, I have updated Figure2-a and Figure2-b into maps in English for all readers to read. See Figures 2-a and Figure2-b on page 4 renew edition.
2. In conclusion, you don't need to list 5.1, 5.2 etc. Instead, please summarize main points of your paper, so that readers can get the primary information / findings from the conclusions.
Response: Thank you for your kind suggestion. I summarize the main points of this article at the end. See the conclusions section from line 307 to line 320 on page 10.
Reviewer 3 Report
Authors try to consider my previous comments but did not achieve it. Carefully check my previous comments and do not reply they were taken into account and no change were made!
Do not use the term heavy metal!
Do not recall words from the title in the keywords list.
Expand your litterature review.
Figures 2a and b are not of good quality and partly in Chinese.
Reference list should be totally rewritten. Citation in the text as well!
It is very difficult to follow science with this lack of rigor of writting.
Author Response
Authors try to consider my previous comments but did not achieve it. Carefully check my previous comments and do not reply they were taken into account and no change were made.
1. Do not use the term heavy metal.
Response: Thanks to the kind suggestion. I have modified the term heavy metal follow according to your comments. For example, the term heavy metal in the article title , line 69 of page 3, line 278 of page 10 were deleted, respectively, as well as line225 the term Hg has been changed into the term mercury.
2.Do not recall words from the title in the keywords list.
Response: Thank you for your kind suggestion. I am very sorry for the words in the title of the article recalled in the keyword list. I have updated the divalent mercury ion into mercury in line 26 of the first page.
3.Expand your literature review.
Response: We thank you very much for your kind advice. The introduction part of this article has been expanded, and the subsequent related research will be further refined.
4.Figures 2a and b are not of good quality and partly in Chinese.
Response: We are very sorry that Figures 2-a and 2-b are not well done, which affects reading. Based on your suggestion, we have updated the map in English for all readers to read.
5.Reference list should be totally rewritten. Citation in the text as well.
Response: We are grateful to for your kind advice. we have rearranged the references to match the Arabic numerals in the text. Besides, three references were added to illustrate in situ restoration of water environment.
6. It is very difficult to follow science with this lack of rigor of writing.
Response: We thank you very much for your suggestion and accept your comments. Follow-up articles should be rigorous and solid.